# Formulation to Calculate Isothermal, Non-Newtonian Elastohydrodynamic Lubrication Problems Using a Pressure Gradient Coordinate System and Its Verification by an Experimental Grease

**Kunihiko Kakoi**

TriboLogics Corporation, Tokyo 110-0016, Japan; kakoi@tribology.co.jp

**Abstract:** This paper presents a formulation of point contact elastohydrodynamic lubrication analysis for an isothermal, non-Newtonian flow. A coordinate system of the pressure gradient was employed herein. A Couette flow and a Poiseuille flow were considered along the directions of the zero and non-zero pressure gradients, respectively. The Poiseuille flow velocity was assumed to be represented by a 4th-order polynomial of z along the film thickness direction. The Couette flow velocity was assumed to be represented by a linear function of z. Subsequently, the modified Reynolds equation, which contains an equivalent viscosity, was obtained. Using Bauer's rheological model, the formulation presented in this study was applied to a grease that has been previously experimented upon. The results of previous studies were compared with those of the present study and a reasonable agreement was noted. The distribution of the equivalent viscosity showed a notable difference from that of Newtonian flow. The formulation can be incorporated easily to the usual elastohydrodynamic lubrication calculation procedure for Newtonian flow. The method can be easily applied to other non-Newtonian rheological models. The equivalent viscosity can be calculated using the one-parameter Newton-Raphson's method; as a result, the calculation can be performed within a reasonable time.

**Keywords:** elastohydrodynamic lubrication; isothermal; non-Newtonian; point contact; grease lubrication; Bauer's model; pressure gradient; equivalent viscosity

## 1. Introduction

Performing experiments on non-Newtonian flows, including grease flows, is considerably time-consuming and costly. Therefore, it is important to numerically analyze the phenomena corresponding to non-Newtonian flows. Numerical approaches can help obtain a variety of data that cannot be determined experimentally. Grease flows can be well defined using Bauer's model; however, owing to the extreme complexity of this model, it is difficult to determine the exact solution as well as approximate solutions for point contact, isothermal, non-Newtonian elastohydrodynamic lubrication (EHL) analyses. Therefore, it is convenient that the non-Newtonian EHL calculation can be executed within a reasonable calculation time and without large modification to the usual Newtonian EHL calculation procedure. Kochi et al. [1] performed experiments on grease under soft EHL conditions and measured the film thickness and traction forces. The method proposed in the present study can be applied to the grease considered in Kochi et al. [1] to validate this theoretical approach.

### 1.1. Classification of Calculation Methods

As shown in Figure 1, the Z-direction is considered to be the film thickness direction. The flow velocities along the X- and Y-directions are denoted by $u$ and $v$, respectively, which are functions of $x$, $y$, and $z$; however, when considering only $z$ dependency, the velocities can be expressed as $u(z)$ and $v(z)$, respectively. The numerical methods for isothermal,

non-Newtonian EHL analyses can be classified in terms of the accuracy of $u(z)$ and $v(z)$, as follows:

Method 1: Exact solution of $u(z)$ and $v(z)$ is obtained.
Method 2: Approximate solution of $u(z)$ and $v(z)$ is obtained.

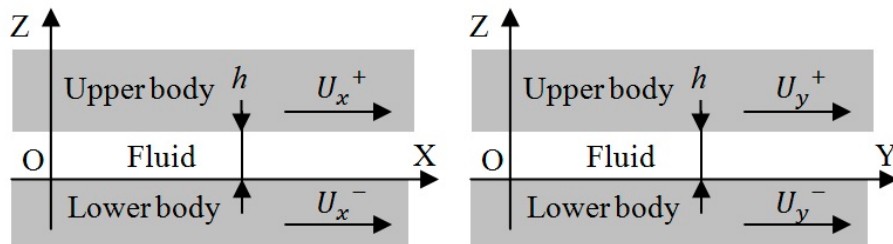

**Figure 1.** Global coordinate system O, X, Y, Z.

Method 2 can be further classified in terms of the employed coordinate system.

Method 2-A: Local X-direction is the sliding direction.
Method 2-B: Local X-direction is the direction of the pressure gradient.

*1.2. Previous Works*

Several methods have been proposed to solve isothermal, non-Newtonian EHL problems. Kauzlarich and Greenwood [2] obtained the exact solution of line contact EHL problems considering the Herschel–Bulkley model. Conry et al. [3] obtained an exact solution of line contact EHL problems by considering Eyring's model. Specifically, the velocity $u(z)$ was represented by a function containing cosh, indicating that, in general, $u(z)$ and $v(z)$ cannot be precisely represented using polynomials of $z$. Dong and Qian [4] obtained an approximate solution of line contact EHL problems considering Bauer's model and using the weighted residual method. Peiran and Shizhu [5] proposed a method to obtain an exact solution of point contact EHL problems for general rheological models by using an equally divided Z-direction mesh. Subsequently, the authors applied the method to a line contact EHL problem. Kim et al. [6] attempted to obtain an exact solution of point contact EHL problems by considering Eyring's model. Ehret et al. [7] considered the X-direction to align with the sliding direction and obtained an approximate solution of $u(z)$ and $v(z)$ represented by the 2nd order polynomial of $z$ by using the perturbation method. Thus, researchers have obtained two effective viscosities: one along the sliding direction and another along the perpendicular direction.

Greenwood [8] focused on the considerable amount of computation time required to obtain an exact solution and compared two approximation methods. Sharif et al. [9] considered the X-direction to align with the sliding direction and developed a method to obtain an exact solution of point contact EHL problems for an arbitrary rheological model. Using this approach, the authors obtained two effective viscosities along the X- and Y-directions. Liu et al. [10] formulated a method to obtain an exact solution of point contact thermal EHL problems considering Eyring's model. Yang et al. [11] formulated a general Reynolds equation for point contact EHL problems by dividing the flow into Couette and Poiseuille flows. Subsequently, the authors obtained an exact solution of line contact EHL problems considering the power law model and demonstrated the effectiveness of their proposed method. Furthermore, Bordenet et al. [12] obtained an exact solution of pure rolling point contact EHL problems considering Bauer's model for $n = 1/2$ and applied the approach to grease.

## 2. Overview of the Proposed Method

In this work, an isothermal, non-Newtonian EHL formulation considering Method 2-B was developed. Although this approach does not yield the exact solution of $u(z)$ and $v(z)$, the calculation is simple and fast. As shown in Figure 2, the local Xc-direction is

considered as the direction of the pressure gradient *d*, and the Yc-direction is considered to be perpendicular to Xc. The flow velocities in the Xc- and Yc-directions are denoted as $u_c$ and $v_c$, respectively. As the pressure gradient toward Yc is zero, the flow $v_c$ is assumed to be a Couette flow, which can be represented using a linear function of *z*. As the pressure gradient toward Xc is generally non-zero, the flow $u_c$ is assumed to be a Poiseuille flow, which can be represented using a 4th-order polynomial of *z*. As $u_c$ and $v_c$ cannot be precisely represented by polynomials of *z*, they are expanded using polynomials of *z*. In such cases, the 6th-order or even higher order polynomials can be considered; however, in this work, a lower 4th-order polynomial was employed. A viscosity corresponding to the Newtonian flow was obtained and termed as the equivalent viscosity. To replace the viscosity of the Newtonian flow with the equivalent viscosity, which is a typical process when evaluating EHL problems, the method proposed by Venner and Lubrecht [13] can be used without any change for the isothermal, non-Newtonian EHL calculation. Given a rheological model, any non-Newtonian isothermal point contact EHL problem can be solved using the proposed method.

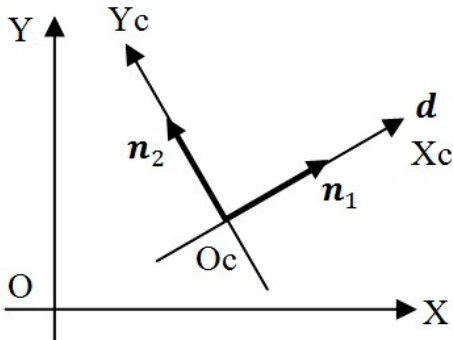

**Figure 2.** Local coordinate system Oc, Xc, Yc.

The calculation procedure consists of two steps. First, the Newtonian EHL calculation is executed for the base oil that yields the Newtonian pressure distribution. Second, using the pressure distribution as the initial value, the non-Newtonian EHL calculation is executed for the non-Newtonian flow. The local coordinate system, Oc, Xc, and Yc, at a particular point, depends on the pressure gradient and it is determined simultaneously in the process to obtain the pressure distribution. The equivalent viscosity is calculated based on the local coordinate system in each iteration loop to obtain the pressure distribution. Here, the method was applied to an experimental grease characterized by Bauer's model.

### 3. Calculation of Velocity Distribution as a Function of *z*

Figure 1 presents the XYZ coordinate system. The force balance of a fluid can be expressed as follows [10]:

$$\frac{\partial \tau_x}{\partial z} = \frac{\partial P}{\partial x} \tag{1}$$

$$\frac{\partial \tau_y}{\partial z} = \frac{\partial P}{\partial y} \tag{2}$$

where *P* is the pressure of the fluid, and $\tau_x$ and $\tau_y$ are the shear stresses in the X- and Y-directions, respectively. The parameters $\tau$ and $\dot{\gamma}$ are defined as follows:

$$\tau = \sqrt{\tau_x{}^2 + \tau_y{}^2} \tag{3}$$

$$\dot{\gamma} = \sqrt{\frac{\partial u}{\partial z}^2 + \frac{\partial v}{\partial z}^2} \tag{4}$$

In Bauer's model, $\tau$ is assumed to be represented as a function of $\dot{\gamma}$, as follows [1,14,15]:

$$\tau = \left(\tau_0 + k_1 \cdot \dot{\gamma} + k_2 \cdot \dot{\gamma}^n\right) \cdot \frac{\eta_n}{\eta_0} \tag{5}$$

Here, $\tau_0$, $k_1$, $k_2$, and $n$ are Bauer's rheological parameters, and $\eta_n$ and $\eta_0$ represent the $P$ dependent viscosity and ambient viscosity of the base oil, respectively. In this work, according to Dong and Qian [4], the parameters $\tau_0$, $k_1$, $k_2$, and $n$ were assumed to be $P$-independent known values determined from the $\tau$-$\dot{\gamma}$ curve measured at the ambient pressure. In Eyring's model, according to Conry et al. [3] and Johnson and Tevaarwerk [16], the relationship between $\tau$ and $\dot{\gamma}$ can be expressed as follows:

$$\dot{\gamma} = \frac{\tau_0{}'}{\eta_n} \sinh\left(\frac{\tau}{\tau_0{}'}\right)$$

Here, $\tau_0{}'$ is Eyring's rheological parameter. Therefore, the relationship between $\tau$ and $\dot{\gamma}$ can be rewritten as follows:

$$\tau = \tau_0{}' \cdot \sinh^{-1}\left(\frac{\eta_n \dot{\gamma}}{\tau_0{}'}\right)$$

The effective viscosity $\eta^*$ can be defined as follows:

$$\eta^* = \frac{\tau}{\dot{\gamma}} \tag{6}$$

The effective viscosity $\eta^*$ is a function of $\dot{\gamma}$, which in turn is a function of $z$. The shear stresses $\tau_x$ and $\tau_y$ are assumed to be represented as

$$\tau_x = \eta^* \frac{\partial u}{\partial z} \tag{7}$$

$$\tau_y = \eta^* \frac{\partial v}{\partial z} \tag{8}$$

Substituting Equations (7) and (8) into Equations (1) and (2), respectively, yields

$$\frac{\partial}{\partial z}\left(\eta^* \frac{\partial u}{\partial z}\right) = \frac{\partial P}{\partial x} \tag{9}$$

$$\frac{\partial}{\partial z}\left(\eta^* \frac{\partial v}{\partial z}\right) = \frac{\partial P}{\partial y} \tag{10}$$

The pressure gradient vector $\boldsymbol{d}$ is defined as follows:

$$\boldsymbol{d} = \left(\frac{\partial P}{\partial x}, \frac{\partial P}{\partial y}\right) \tag{11}$$

Similar to the method employed by Yang et al. [11], this method involves the flow being divided into Couette and Poiseuille flows. As shown in Figure 2, the Xc-direction is considered to be along $\boldsymbol{d}$, and its direction vector is $\boldsymbol{n}_1$. The Yc-direction is perpendicular to $\boldsymbol{d}$, and its direction vector is $\boldsymbol{n}_2$. The local coordinate system, Oc, Xc, and Yc, at a given point depends on the pressure gradient and is determined simultaneously in the process to obtain the pressure distribution. In the Xc and Yc coordinate system, Equations (9) and (10) can be rewritten as follows:

$$\frac{\partial}{\partial z}\left(\eta^* \frac{\partial u_c}{\partial z}\right) = \dot{d} \tag{12}$$

$$\frac{\partial}{\partial z}\left(\eta^* \frac{\partial v_c}{\partial z}\right) = 0 \tag{13}$$

$$d = \sqrt{\boldsymbol{d} \cdot \boldsymbol{d}} \tag{14}$$

Furthermore, the velocities $u_c(z)$ and $v_c(z)$ satisfy the following boundary conditions:

$$u_c(0) = U^-, \; u_c(h) = U^+, \; v_c(0) = V^-, \; v_c(h) = V^+ \tag{15}$$

Here, $U^+$ and $U^-$ denote the velocities of the upper and lower surfaces in the Xc-direction, respectively; $V^+$ and $V^-$ denote the velocities of the upper and lower surfaces in the Yc-direction, respectively. Although the velocities $u_c(z)$ and $v_c(z)$ cannot be represented by polynomials exactly [3], here they are approximated and expanded using polynomials of $z$ so that the velocities satisfy Equation (15), as follows. In this case, the variables $a_1$ and $a_2$ are unknown.

$$u_c(z) = \frac{\Delta U}{h} \cdot z + U^- - a_1 \cdot z(h - z) - a_2 \cdot z^2 (h - z)^2 \tag{16}$$

$$v_c(z) = \frac{\Delta V}{h} \cdot z + V^- \tag{17}$$

Here, $h$ is the fluid film thickness, and $\Delta U$ and $\Delta V$ denote the velocity differences; specifically, $\Delta U = U^+ - U^-$ and $\Delta V = V^+ - V^-$. A higher order term of $z$, for example, $z^3(h-z)^3$, can also be considered; however, in this work, the lower-order approximation was chosen. As the pressure gradient toward the Yc direction is zero, $v_c$ was assumed to be a Couette flow and approximated considering a linear equation of $z$. Furthermore, as the pressure gradient toward the Xc-direction is generally non-zero, $u_c$ was assumed to be a Poiseuille flow and approximated using a 4th-order polynomial of $z$. If $d = 0$, then $u_c$ is also a Couette flow, and Equation (16) can be replaced with the following equation:

$$u_c(z) = \frac{\Delta U}{h} \cdot z + U^- \tag{18}$$

The following equations are derived from Equations (16) and (17):

$$\frac{\partial u_c}{\partial z} = u'(z) = \frac{\Delta U}{h} - a_1 \cdot (h - 2z) - a_2 \cdot 2z(h - z)(h - 2z) \tag{19}$$

$$\frac{\partial v_c}{\partial z} = v'(z) = \frac{\Delta V}{h} \tag{20}$$

The following equations are derived from Equation (19):

$$u'(0) = \frac{\Delta U}{h} - a_1 h \tag{21}$$

$$u'(h) = \frac{\Delta U}{h} + a_1 h \tag{22}$$

$$u'\left(\frac{h}{4}\right) = \frac{\Delta U}{h} - a_1 \frac{h}{2} - a_2 \frac{3h^3}{16} \tag{23}$$

$$u'\left(\frac{3h}{4}\right) = \frac{\Delta U}{h} + a_1 \frac{h}{2} + a_2 \frac{3h^3}{16} \tag{24}$$

If rheological constitutive equations are given, the effective viscosity $\eta^*(z)$ can be calculated using Equations (4)–(6), (16) and (17). The integration of Equation (12) from $z = 0$ to $z = h$ yields Equation (25), and the integration of Equation (12) from $z = 3h/4$ to $z = 3h/4$ yields Equation (26). These equations are used to determine the values of $a_1$ and $a_2$.

$$\eta^*(h) \cdot \left(\frac{\Delta U}{h} + a_1 h\right) - \eta^*(0) \cdot \left(\frac{\Delta U}{h} - a_1 h\right) = dh \tag{25}$$

$$\eta^*\left(\frac{3h}{4}\right)\cdot\left(\frac{\Delta U}{h}+a_1\frac{h}{2}+a_2\frac{3h^3}{16}\right)-\eta^*\left(\frac{h}{4}\right)\cdot\left(\frac{\Delta U}{h}-a_1\frac{h}{2}-a_2\frac{3h^3}{16}\right)=\frac{dh}{2} \quad (26)$$

Equations (21) and (22) show that both $\eta^*(h)$ and $\eta^*(0)$ do not contain $a_2$. Therefore, Equation (25) does not contain $a_2$ and contains only the unknown variable $a_1$. The equation can be solved using the one-variable Newton–Raphson method. Although Equation (26) contains both $a_1$ and $a_2$, $a_1$ has been determined using Equation (25). Consequently, Equation (26) can be considered as an equation involving only the unknown variable $a_2$. Thus, it can also be solved using the one-variable Newton–Raphson method. To determine $a_1$, a non-dimensional variable $b_1$ defined using Equation (27) and a function $f_1(b_1)$ defined using Equation (28) are introduced. The value of $b_1$ can be calculated considering $f_1(b_1)=0$.

$$b_1=\log\left(\frac{2a_1\eta_n}{d}\right) \quad (27)$$

$$f_1(b_1)=\eta^*(h)\cdot\left(\frac{\Delta U}{h}+e^{b_1}\frac{dh}{2\eta_n}\right)-\eta^*(0)\cdot\left(\frac{\Delta U}{h}-e^{b_1}\frac{dh}{2\eta_n}\right)-d\cdot h \quad (28)$$

When $b_1$ is near the solution, $\Delta b_1$ can be calculated using the following equation:

$$0=f_1(b_1+\Delta b_1)\fallingdotseq f_1(b_1)+\frac{df_1}{db_1}\cdot\Delta b_1\fallingdotseq f_1(b_1)+e^{b_1}\frac{dh}{2\eta_n}[\eta^*(h)+\eta^*(0)]\cdot\Delta b_1 \quad (29)$$

In other words, the new candidate $b_{1new}$ of $b_1$ is calculated using the iterative process of Newton–Raphson's method, as follows:

$$b_{1new}=b_1+\Delta b_1=b_1-\frac{f_1(b_1)}{a_1h\,[\eta^*(h)+\eta^*(0)]} \quad (30)$$

As $\eta^*(h)$ and $\eta^*(0)$ are originally functions of $b_1$, $df_1/db_1$ includes $d\eta^*/db_1$; however, in this work, the dependency was ignored, and $\Delta b_1$ was approximated as in Equation (30). To determine $a_2$, a non-dimensional variable $b_2$ defined using Equation (31) and a function $f_2(b_2)$ defined using Equation (32) are introduced. The value of $b_2$ can be calculated considering $f_2(b_2)=0$.

$$b_2=\frac{2a_2h^2\eta_n}{5d} \quad (31)$$

$$f_2(b_2)=\eta^*\left(\frac{3h}{4}\right)\cdot\left(\frac{\Delta U}{h}+a_1\frac{h}{2}+a_2\frac{3h^3}{16}\right)-\eta^*\left(\frac{h}{4}\right)\cdot\left(\frac{\Delta U}{h}-a_1\frac{h}{2}-a_2\frac{3h^3}{16}\right)-\frac{dh}{2} \quad (32)$$

When $b_2$ is near the solution, $\Delta b_2$ can be calculated using the following equation:

$$0=f_2(b_2+\Delta b_2)\fallingdotseq f_2(b_2)+\frac{df_2}{db_2}\cdot\Delta b_2\fallingdotseq f_2(b_2)+\frac{15dh}{32\eta_n}\left[\eta^*\left(\frac{3h}{4}\right)+\eta^*\left(\frac{h}{4}\right)\right]\cdot\Delta b_2 \quad (33)$$

In other words, the new candidate $b_{2new}$ of $b_2$ is calculated using the iterative process of Newton–Raphson's method, as follows:

$$b_{2new}=b_2+\Delta b_2=b_2-\frac{f_2(b_2)}{[\eta^*(3h/4)+\eta^*(h/4)]}\cdot\frac{32\eta_n}{15dh} \quad (34)$$

As $\eta^*(3h/4)$ and $\eta^*(h/4)$ are originally functions of $b_2$, $df_2/db_2$ includes $d\eta^*/db_2$; however, in this work, the dependency was ignored, and $\Delta b_2$ was approximated as in Equation (34). Subsequently, in the iteration process of Newton–Raphson's method, only $\eta^*$ depends on the rheological characteristics. Therefore, if the rheological equation corresponding to Equation (5) is incorporated, any isothermal, non-Newtonian EHL calculation can be performed. As per the Newton–Raphson method, the initial value for both $b_1$ and $b_2$ can be zero. As both variables $a_1$ and $a_2$ are solved using the one-variable Newton–Raphson method, the calculation can be performed within a reasonable time.

## 4. Calculation of Equivalent Viscosity, Flow, and Surface Force

The flow $q_1$ along the $\boldsymbol{n}_1$ direction can be defined using Equation (16), as follows. The density $\rho$ is assumed to be independent of $z$.

$$
\begin{aligned}
q_1 = \int_0^h \rho u_{\rm c}\, dz = \int_0^h \rho \left[ \frac{\Delta U}{h}\cdot z + U^- - a_1 \cdot z(h-z) - a_2 \cdot z^2 (h-z)^2 \right] dz \\
= \rho U h - \frac{\rho a_1 h^3}{6} - \frac{\rho a_2 h^5}{30}
\end{aligned}
\tag{35}
$$

Here, $U$ is the average velocity in the Xc-direction and can be expressed as follows:

$$
U = \frac{U^+ + U^-}{2}
\tag{36}
$$

The equivalent viscosity $\eta_{eq}$ is defined as follows:

$$
\eta_{eq} = \frac{5d}{2(5a_1 + a_2 h^2)}
\tag{37}
$$

Consequently, $q_1$ can be represented as

$$
q_1 = \rho U h - \frac{\rho h^3}{12\eta_{eq}} \cdot d
\tag{38}
$$

The flow $q_2$ along the $\boldsymbol{n}_2$ direction can be derived from Equation (17), as follows:

$$
q_2 = \int_0^h \rho v_{\rm c}\, dz = \rho V h
\tag{39}
$$

Here, $V$ is the average velocity in the Yc direction and can be expressed as follows:

$$
V = \frac{V^+ + V^-}{2}
\tag{40}
$$

Hence, in the XYZ coordinate system, the flow vector $\boldsymbol{q}$ can be expressed as

$$
\begin{aligned}
\boldsymbol{q} &= \left( \rho U h - \frac{\rho h^3}{12\eta_{eq}} \cdot d \right) \cdot \boldsymbol{n}_1 + \rho V h \cdot \boldsymbol{n}_2 \\
&= \rho U h \cdot \boldsymbol{n}_1 + \rho V h \cdot \boldsymbol{n}_2 - \frac{\rho h^3}{12\eta_{eq}} \cdot d \cdot \boldsymbol{n}_1 \\
&= \rho \boldsymbol{U} h - \frac{\rho h^3}{12\eta_{eq}} \cdot \left( \frac{\partial P}{\partial x},\ \frac{\partial P}{\partial y} \right)
\end{aligned}
\tag{41}
$$

Here, $\boldsymbol{U}$ is the average velocity vector of the upper and lower surfaces, defined as follows:

$$
\boldsymbol{U} = U \cdot \boldsymbol{n}_1 + V \cdot \boldsymbol{n}_2 = (U_x,\ U_y)
\tag{42}
$$

$U_x$ and $U_y$ denote the average velocities in the upper and lower surfaces in the XY-direction, respectively. When the mass conservation law is applied to Equation (41), the following modified Reynolds equation is obtained.

$$
\frac{\partial \rho U_x h}{\partial x} + \frac{\partial \rho U_y h}{\partial y} - \frac{\partial}{\partial x}\left( \frac{\rho h^3}{12\eta_{eq}} \cdot \frac{\partial P}{\partial x} \right) - \frac{\partial}{\partial y}\left( \frac{\rho h^3}{12\eta_{eq}} \cdot \frac{\partial P}{\partial y} \right) = 0
\tag{43}
$$

The difference in the representation of Equations (41) and (43) and that of Newtonian flow only pertains to the viscosities $\eta_{eq}$ and $\eta_{\rm n}$, respectively. In fact, the equivalent viscosity $\eta_{eq}$ defined by Equation (37) was determined so that Equations (41) and (43) maintain the same form as that of Newtonian flow. Therefore, the EHL calculation procedure for Newtonian flows, such as the method proposed by Venner and Lubrecht [13], can be

applied to the current calculation by simply replacing $\eta_n$ with $\eta_{eq}$. The shear stress $\tau_1$ along the Xc-direction can be expressed as

$$\tau_1(z) = \eta^* \frac{\partial u_c}{\partial z} = \eta^* \left[ \frac{\Delta U}{h} - a_1 \cdot (h - 2z) - a_2 \cdot 2z(h - z)(h - 2z) \right] \tag{44}$$

Therefore, the surface forces $P_{10}$ and $P_{1h}$ acting on the lower and upper surfaces along the Xc-direction, respectively, can be defined as

$$P_{10} = \tau_1(0) = \eta^*(0) \cdot \left( \frac{\Delta U}{h} - a_1 h \right) \tag{45}$$

$$P_{1h} = -\tau_1(h) = -\eta^*(h) \cdot \left( \frac{\Delta U}{h} + a_1 h \right) \tag{46}$$

The shear stress $\tau_2$ along the Yc-direction is expressed as

$$\tau_2 = \eta^* \frac{\partial v_c}{\partial z} = \eta^* \frac{\Delta V}{h} \tag{47}$$

Therefore, the surface forces $P_{20}$ and $P_{2h}$ acting on the lower and upper surfaces along the Yc-direction, respectively, can be defined as

$$P_{20} = \tau_2(0) = \eta^*(0) \cdot \frac{\Delta V}{h} \tag{48}$$

$$P_{2h} = -\tau_2(h) = -\eta^*(h) \cdot \frac{\Delta V}{h} \tag{49}$$

In the XYZ coordinate system, the surface force vectors $\boldsymbol{P}_0$ and $\boldsymbol{P}_h$ that act on the lower and upper surfaces, respectively, can be expressed as

$$\boldsymbol{P}_0 = P_{10} \cdot \boldsymbol{n}_1 + P_{20} \cdot \boldsymbol{n}_2 \tag{50}$$

$$\boldsymbol{P}_h = P_{1h} \cdot \boldsymbol{n}_1 + P_{2h} \cdot \boldsymbol{n}_2 \tag{51}$$

## 5. Application to a Grease

As mentioned previously, Kochi et al. [1] conducted experiments on grease under soft EHL conditions and measured the film thickness and traction forces. In the present study, the proposed method was applied to one of the greases considered in the study by Kochi et al. [1] so as to validate the theoretical approach. Grease A in the literature [1] was chosen to test the proposed method. The modified Reynolds equation, as expressed in Equation (43), was solved using a multi-level method, as reported by Venner and Lubrecht [13]. The commercial program Tribology Engineering Dynamics Contact Problem Analyzer (TED/CPA) V852 was employed. Figure 3 illustrates the calculation conditions. The upper body was a steel ball, and the lower body was a disk composed of glass or polycarbonate (PC). The rheological properties of the grease were assumed to be represented by Bauer's model, according to existing literature [1]. Detailed properties of the steel, glass, PC, and grease are described in the previous study [1]. The pressure dependency of the density was defined using Dowson–Higginson's formula, as follows:

$$\rho(P) = \rho_0 \cdot \frac{p_0 + \beta \cdot P}{p_0 + P}, \ \rho_0 = 1, \ p_0 = 590 \text{ MPa}, \ \beta = 1.34 \tag{52}$$

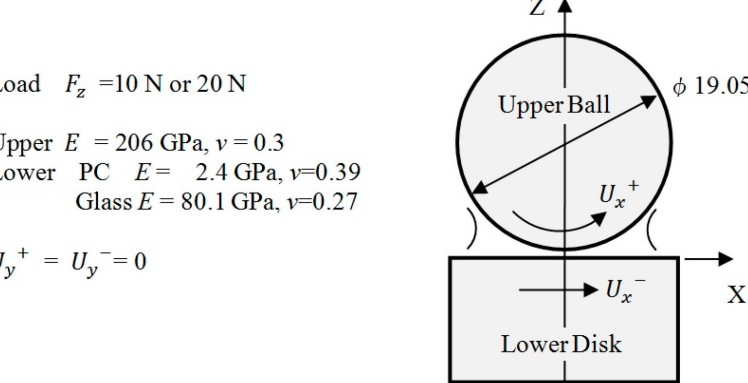

**Figure 3.** Calculation condition.

The pressure dependency of the base oil viscosity was assumed to be defined using Barus' formula:

$$\eta_n(P) = \eta_0 \cdot \exp(\alpha P), \quad \eta_0 : 49.5 \text{ mPa·s}, \quad \alpha : 14 \text{ GPa}^{-1} \tag{53}$$

In particular, Equation (6) diverges when $\dot{\gamma}$ approaches zero. It was assumed that if $\dot{\gamma}$ is lower than a certain value $c_{min}=100.0$ s$^{-1}$, then $\eta^*$ varies linearly with the gradient $d\eta^*/d\dot{\gamma}$ at $c_{min}$. Bauer's parameter $k_1$ was assumed to be the base oil ambient viscosity $\eta_0$. The values of Bauer's parameters $\tau_0$, $k_2$, and $n$ were determined from the apparent viscosity curve of grease A, as shown in Figure 9 of Kochi et al. [1]. When $P = 0$, the curve was assumed to pass through the following three points: 100 mm/s, 6.46475 Pa·s; 10,000 mm/s, 0.16712 Pa·s; and 1,000,000 mm/s, 0.06573 Pa·s.

The values of $\tau_0$, $k_2$, and $n$ were determined to ensure that Bauer's curve passes through the abovementioned three points, as follows:

$$\tau_0 = 0.000621839 \text{ MPa}, \quad k_2 = 6.99118 \cdot 10^{-7}, \quad n = 0.7248 \tag{54}$$

Figure 4 presents the central film thickness of grease A and base oil as a function of the rolling velocity in the case of pure rolling and $F_z = 10$ N. The solid lines show the calculation results and the dotted lines show the experimental results. The experimental data were read using the caliper from Figure 4 of Kochi et al. [1]. Figure 4a,b shows the cases of a PC disk and glass disk, respectively. The calculation range was set as follows: $-1.0 \leq X \leq 0.4$ and $-0.6 \leq Y \leq 0.6$.

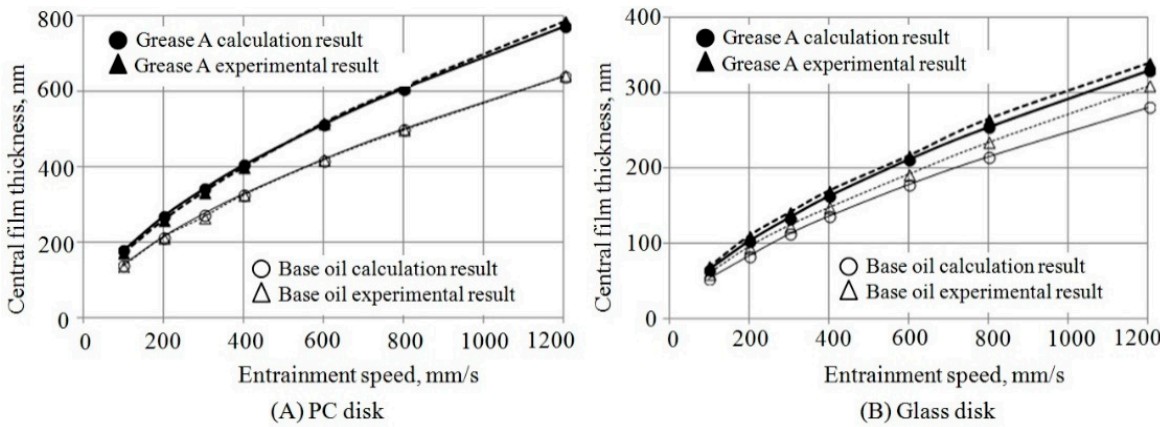

**Figure 4.** Comparison of film thickness between the calculation and the experiment.

However, in the case of grease A, a glass disk, and a velocity of less than or equal to 300 mm/s, the range was set as follows: $-0.35 \leq X \leq 0.14$ and $-0.2 \leq Y \leq 0.2$.

In these cases, the oil film was thin and the wide range calculation became hard to perform. In the case of the PC disk, the calculation results exhibited good agreement with the experimental results. In the case of the glass disk, the results of the base oil showed some difference but the other data showed reasonable agreement. Figure 5, which illustrates a sample calculation, shows the distribution of $P$, $h$, and $\eta_{eq}$ for the case involving a PC disk, a pure rolling velocity of 1200 mm/s, and grease A. Typically, in the case of pure rolling velocity, $\dot\gamma$ is small and $\eta_{eq}$ is large. In addition, only the appearance of the distribution of $\eta_{eq}$ differs from that of the base oil. Figure 5d shows the distribution of $\eta_{eq}$ at section Y = 0. It can be seen that $\eta_{eq}$ becomes extremely large at the center, where the pressure gradient is small and the flow volume is low.

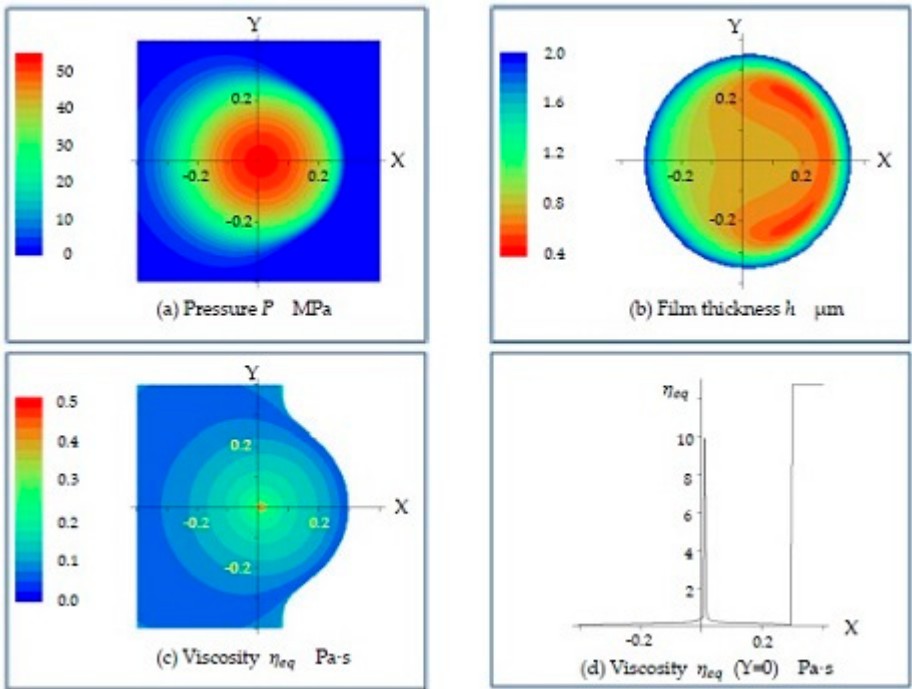

**Figure 5.** Distribution of $P$, $h$, and $\eta_{eq}$ of grease A at a rolling velocity of 1200 mm/s.

Figure 6 presents the traction coefficient as a function of the slide roll ratio when $F_z = 20$ N. The solid lines show the calculation results and the dotted lines show the experimental results. The experimental data were read using the caliper from Figure 8 of Kochi et al. [1]. The slide roll ratio is the difference between the upper and lower velocities divided by their average value. The traction coefficient was calculated according to the approaches proposed in the existing literature [1,17]:

$$TRC = \frac{TX0 - TXh}{2F_z} \tag{55}$$

Here, $TX0$ and $TXh$ denote the X-direction traction forces acting on the lower and upper surfaces, respectively; $F_z$ is the load. The calculation range was $-0.8 \leq X \leq 0.5$, $-0.5 \leq Y \leq 0.5$. The calculation results were in fairly good agreement with the experimental results.

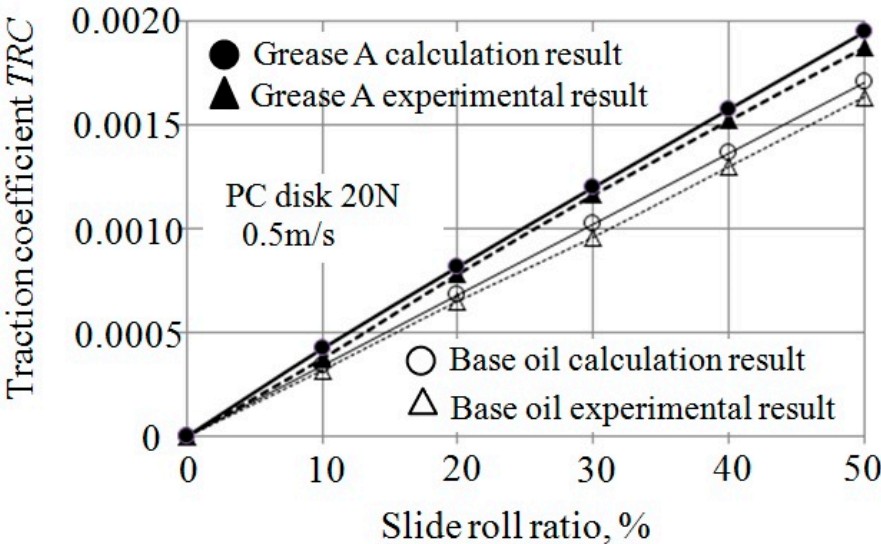

**Figure 6.** Comparison of the traction coefficient between the calculation and the experiment.

Figure 7, which illustrates a sample calculation, shows the distribution of $P$, $h$, and $\eta_{eq}$ for the case involving grease A and a slide roll ratio of 10%. Only the appearance of the distribution of $\eta_{eq}$ differs from that of the base oil, exhibiting a figure eight in the vicinity of the contact point. Figure 7c,d shows the same distribution of $\eta_{eq}$ in different display ranges. It can be observed that $\eta_{eq}$ reduces in the rapid flow region. The figure eight phenomenon is characteristic of non-Newtonian flow, in which the apparent viscosity becomes large at a low velocity gradient. This phenomenon can be explained as follows. Let the XY coordinates of points A, B, C, and D be $(-0.1, 0)$, $(+0.1, 0)$, $(0, -0.1)$, and $(0, +0.1)$, respectively. The velocity gradient vector $\boldsymbol{n}_1$ at these points are approximately $(+1, 0)$, $(-1, 0)$, $(0, +1)$, and $(0, -1)$, respectively, as shown in Figure 8a. When $a_2$ in Equation (37) is neglected, $\eta_{eq}$ can be expressed as follows:

$$\eta_{eq} = \frac{d}{2a_1} \tag{56}$$

Calculating $a_1$ from Equation (25) and substituting it into Equation (56) yields:

$$\eta_{eq} = \frac{[\eta^*(h) + \eta^*(0)]}{2} \cdot \frac{1}{1 + \frac{\Delta U}{dh^2}[\eta^*(0) - \eta^*(h)]} \tag{57}$$

At points C and D, the direction of flow and that of $\boldsymbol{n}_1$ is orthogonal, so $\Delta U = 0$. Consequently, the equivalent viscosity parameters $\eta_{eq,C}$ and $\eta_{eq,D}$ at points C and D are given as follows:

$$\eta_{eq,C} = \eta_{eq,D} = \frac{[\eta^*(h) + \eta^*(0)]}{2} \tag{58}$$

At point A, where $\boldsymbol{n}_1$ directs towards $+$X and the velocity gradient at $Z = h$ is greater than that at $Z = 0$ (as shown in Figure 8b), the following equation is satisfied:

$$\eta^*(0) > \eta^*(h), \ \Delta U = U_x{}^+ - U_x{}^- > 0 \tag{59}$$

At point B, where $\boldsymbol{n}_1$ directs towards $-$X, and the velocity gradient at $Z = h$ is smaller than that at $Z = 0$ (as shown in Figure 8c), the following equation is satisfied:

$$\eta^*(0) < \eta^*(h), \ \Delta U = -\left(U_x{}^+ - U_x{}^-\right) < 0 \tag{60}$$

In any case, the equivalent viscosity parameters $\eta_{eq,A}$ and $\eta_{eq,B}$ at points A and B become smaller than $\eta_{eq,C}$ and $\eta_{eq,D}$ by the effect of the second term of Equation (57). In the

pure rolling case, where $\Delta U$ is 0, Equation (57) results in Equation (58). It can be understood that in such a case, no figure-eight-shaped distribution appears as is shown in Figure 5c.

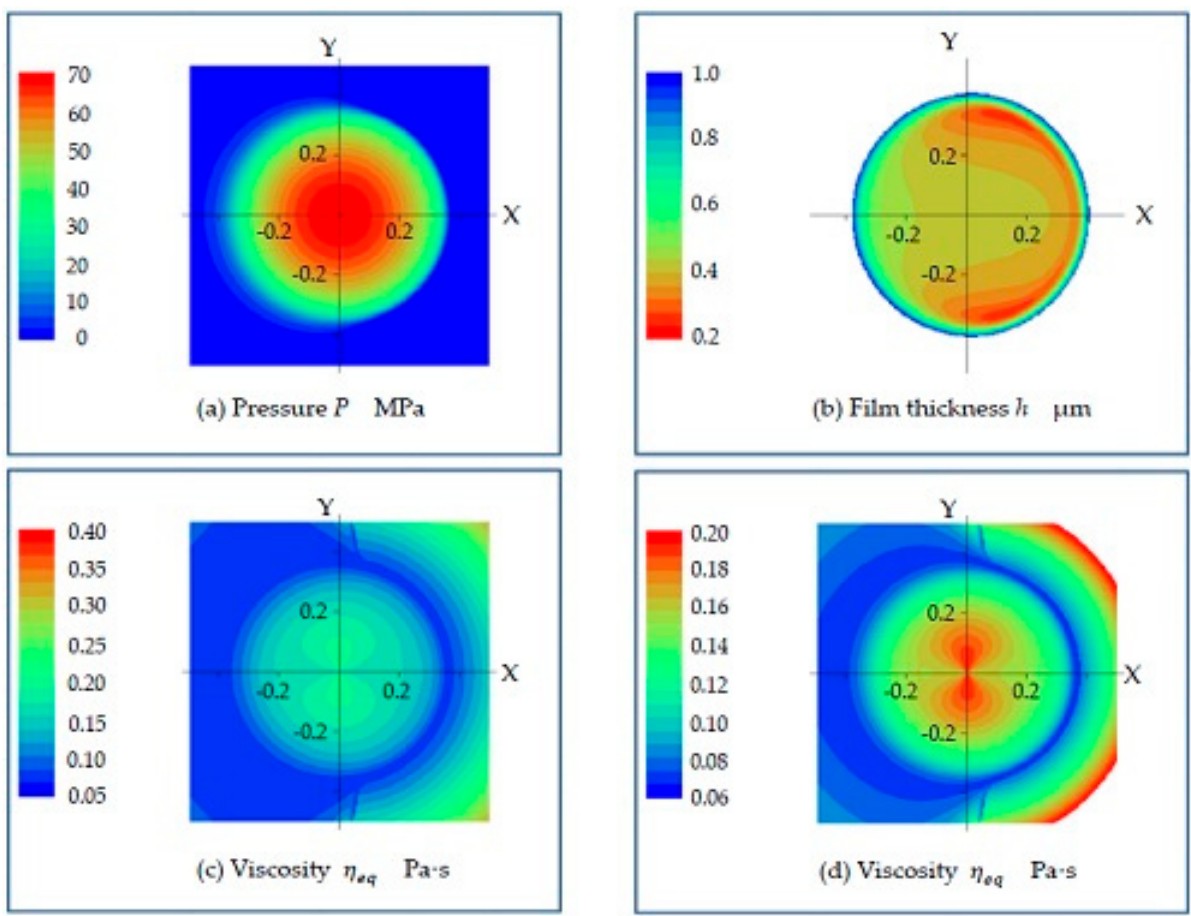

**Figure 7.** Calculation results of grease A at a 10% slide ratio.

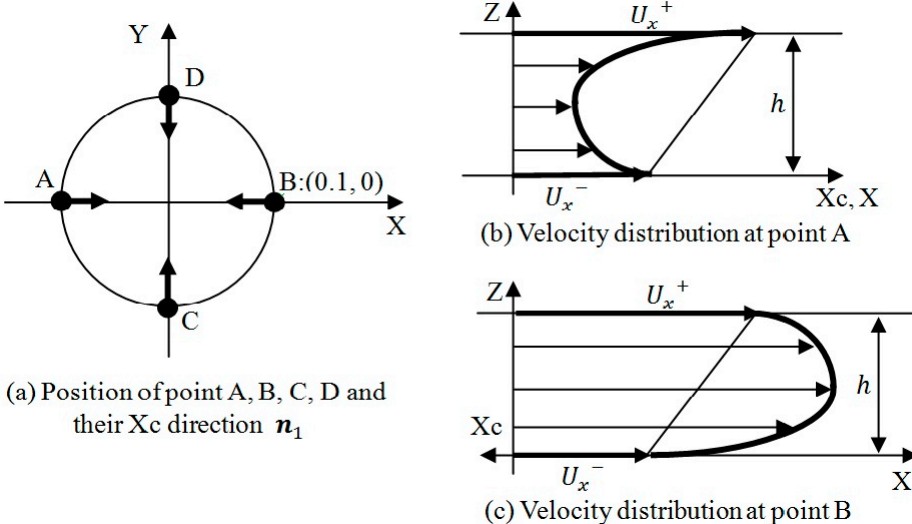

**Figure 8.** Explanation of the figure eight phenomenon.

## 6. Conclusions

In this study, an isothermal, non-Newtonian EHL formulation of Bauer's model was performed using the local coordinate system of the pressure gradient. The flow toward

the pressure gradient was assumed to be a Poiseuille flow and was approximated using a 4th-order polynomial of $z$. The flow along the direction of the zero pressure gradient was assumed to be a Couette flow and was approximated using a linear function of $z$. The following results were obtained.

(1) A modified Reynolds equation, which contains an equivalent viscosity, was obtained.
(2) The EHL calculation procedure for Newtonian flows can be applied to non-Newtonian flows by simply replacing the Newtonian viscosity with the equivalent viscosity.
(3) If rheological equations are incorporated, any isothermal, non-Newtonian EHL calculation can be performed easily.
(4) As the equivalent viscosity is calculated using the one-variable Newton–Raphson method, the EHL calculation can be performed within a reasonable calculation time.
(5) Using Bauer's model, the formulation was applied to a grease that was evaluated experimentally by Kochi et al. [1]. The results obtained using the proposed method and the experimental results were compared, and reasonable agreement was noted.
(6) In the case of sliding velocity, the equivalent viscosity shows a figure-eight-shaped distribution in the vicinity of the contact point.

However, the proposed method yields an approximate solution. If the Poiseuille flow and Couette flow cannot be approximated using a 4th-order polynomial of $z$ and a linear function of $z$, respectively, the obtained results may be inaccurate. The application limits of the current formulation are not clear. Therefore, future work should be focused on determining these limitations.

**Funding:** This research was funded by the TriboLogics Corporation.

**Acknowledgments:** The author sincerely thanks TriboLogics Corporation (www.tribology.co.jp/indexEng.htm, accessed on 13 May 2021) for permitting the use of TED/CPA. The author is also very grateful to Hiroyuki OHTA of the Nagaoka University of Technology, Japan, for his constant guidance, encouragement, and valuable suggestions.

**Conflicts of Interest:** The author declares no conflict of interest.

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
