# Peer review of "Formulation to Calculate Isothermal, Non-Newtonian Elastohydrodynamic Lubrication Problems Using a Pressure Gradient Coordinate System and Its Verification by an Experimental Grease"

_lubricants, doi:10.3390/lubricants9050056_

Round 1

Reviewer 1 Report

Thank you for consideration of my comments. You are not able to answer all questions - but O.k. May be you have to discuss with the audience after publishing

Reviewer 2 Report

Though the new manuscript is better than the original one and some of the issues/concerns were addressed by the author, the essence of this work is still the same with the same shortcomings. Most of the critical points are still not addressed satisfactorily. Therefore, the manuscript is still not acceptable for publication in Lubricants.

Reviewer 3 Report

The paper introduces a formulation of an isothermal, non-Newtonian Couette (a linear function of z) and Poisseuille (4th-order polynomial of z along the film thickness direction) flow and a respective Reynolds equation to be used in the simulation of elastohydrodynamic contacts lubricated by grease. Here, this was done by means of Bauer’s rheological model. The clue is the determination of an equivalent viscosity, which is then included in the Reynolds equation, neglecting viscosity changes in the direction of the lubricant gap height.

The paper places a strong focus on the theoretical aspects, which is well done and interesting. The reader would also wish for more results/discussion so that this is balanced with the theory a bit better. As stated, if the Poiseuille flow and Couette flow cannot be approximated using a 4th order polynomial of z and a linear function of z, respectively, results may be inaccurate and the application limits of the formulation are therefore not clear. However, for publication this should be more mature and the author should expand this discussion significantly. Furthermore, since the focus is actually on the EHL contact, this should also be addressed more in the manuscript, both in the introduction and also in the methods section. What simulation approaches exist in EHL and how can the presented approach be implemented there? How was it implemented here and how does the used EHL simulation work after all (this was not explained in enough detail)? Figures 8-10 are comparatively large and could be shortened to one figure (a-c).

If the points raised have been met, the paper might be considered.

Reviewer 4 Report

Title: 

  • The title reads rather bulky. It should be simplified and shortened.

Keywords: 

  • The number of keywords used/proposed should be reduced.

Abstract: 

  • The abstract should be rewritten. It contains too much general information and not enough take home message. Please revise.

Introduction: 

  • The first part of the introduction falls rather short. It just contains 1 citation. Please largely extend the respective state-of-the art of the paper.
  • Related to numerical solutions of EHL problems, the summarizing article of Marian et al. should be considered entitled as "Non-Dimensional Groups, Film Thickness Equations and Correction Factors for Elastohydrodynamic Lubrication: A Review"
  • The mathematical formulation of the paper should be shortened and streamlined. You should consider to put a significant amount of the equations in an Appendix. This will help to improve the readability of your article. 

- The following statement remains unclear "The proposed method was applied to the same grease considered in the study by Kochi et al. [1] to validate the theoretical approach"... Please verify

  • How many different greases have you studied to validate the respective approach?
  • The captions of the figures should be improved and extended. 
  • In figure 4, experimental results are shown. It remains unclear whether this is the experimental work of the author? If yes, all the experimental details are missing. If no, proper citation and referencing are missing. 
  • The conclusions and take home messages based upon this article are rather limited since it seems that only one grease has been studied. If possible, please extend your work. 
  • Figure 8, 9 and 10 should be combined. 
  • The list of references must be extended. 

Round 2

Reviewer 3 Report

The paper can be accepted since the theoretical considerations could be of interest. As stated before, the reader would wish for more results/discussion so that this is balanced with the theory.  This may therefore not really satisfy expectations of a research article, and the manuscript has more the character of a technical note or similar.

Reviewer 4 Report

Unfortunately, my concerns and questions were not answered. Most of the comments were answered by "I apologize for this oversight", but no changes were made in the manuscript. Just some insignificant changes were made in the manuscript and most of my concerns ignored. The number of citations is still very limited, which does not reflect a proper review of the existing state of the art. 

The mathematical formulation is still the same. No changes have been made. 

The quality of the manuscript has not substantially improved after this revision round. 

This manuscript is a resubmission of an earlier submission. The following is a list of the peer review reports and author responses from that submission.

Round 1

Reviewer 1 Report

The current work presents the development of a modified Reynolds equation based on Bauer’s rheological model for greases. The formulation assumes a Poiseuille flow in the x-direction and Couette flow in the y-direction, which is a rather strange choice, as it limits the application to line contacts that are rolling in the x-direction and sliding in the y-direction, for which this reviewer cannot really figure out the engineering application. But, then in the results section, the author employs this formulation to study point contacts where there would be a Poiseuille component in all space directions, which is even more strange, since the derived Reynolds equation is not applicable in that case. Besides, the main novelty of this work is in the adoption of a 4th-order polynomial approximation of the Poiseuille velocity components, which in my opinion is unnecessary, and the commonly adopted 2nd-order approximation in the literature would have been sufficient. This is confirmed in the results section where little difference is revealed in terms of both film thickness and friction predictions between the two approximations. Also, in the results section the author mentions experimental results and a good agreement between the reported predictions and those results, but the experimental data is not shown.  Some of the reported results also show some strange features e.g. fig5 (d) shows an unexpected rise in the effective viscosity of the grease towards the contact outlet. The interpretation of the physics behind the 8-shape in fig7 (d) also does not make sense to this reviewer. With all the above being said, I am afraid that the current work cannot be accepted for publication in Lubricants.

Reviewer 2 Report

Thank you very much for your interesting research work. Some comments from my point of view:

  • it seems you select the Bauer's model because of the Ref[1].Can you give a brief assessment of whether it is the most suitable model. Because there are a number of other models for lubricating grease
  • Please can you give some rheological properties of the grease A to make it more clear what type of grease you investigate.
  • Why did you named the sample Grease A? Because you investigate only one.
  • I did not understand what you mean with Fig.4.All curves come from [1]? If not please mark
  • The same with Fig.6.
  • It is well known that greases show itis known that lubricating greases show a strongly time-dependent behavior. Can you give a sentence about this phenomenon and your developed model
  • From my point of view the conclusion is to short. Please can you give some more aspect from you work. For example its not clear why you applicate your model to greases.

Reviewer 3 Report

Dear Author

Figures 1 & 2 need to be improved in order to illustrate the three dimensions more clear. As an example in Figure 1 you mention Y coordinate but this is not illustrated in the figure. Additionally please explain the error in this methodology.